# Reliability of citations of medRxiv preprints in articles published on COVID-19 in the world leading medical journals

Jean-Francois Gehanno[1,2]*, Julien Grosjean[2,3], Stefan J. Darmoni[2,3], Laetitia Rollin[1,2]

1 Department of Occupational Medicine, Rouen University Hospital, Rouen, France, 2 Inserm, Rouen University, Sorbonne University, University of Paris 13, Laboratory of Medical Informatics and Knowledge Engineering in e-Health, LIMICS, Paris, France, 3 Department of Biomedical Informatics, Rouen University Hospital, Rouen France

* jf.gehanno@chu-rouen.fr

## Abstract

### Introduction

Preprints have been widely cited during the COVID-19 pandemics, even in the major medical journals. However, since subsequent publication of preprint is not always mentioned in preprint repositories, some may be inappropriately cited or quoted. Our objectives were to assess the reliability of preprint citations in articles on COVID-19, to the rate of publication of preprints cited in these articles and to compare, if relevant, the content of the preprints to their published version.

### Methods

Articles published on COVID in 2020 in the BMJ, The Lancet, the JAMA and the NEJM were manually screened to identify all articles citing at least one preprint from medRxiv. We searched PubMed, Google and Google Scholar to assess if the preprint had been published in a peer-reviewed journal, and when. Published articles were screened to assess if the title, data or conclusions were identical to the preprint version.

### Results

Among the 205 research articles on COVID published by the four major medical journals in 2020, 60 (29.3%) cited at least one medRxiv preprint. Among the 182 preprints cited, 124 were published in a peer-reviewed journal, with 51 (41.1%) before the citing article was published online and 73 (58.9%) later. There were differences in the title, the data or the conclusion between the preprint cited and the published version for nearly half of them. MedRxiv did not mentioned the publication for 53 (42.7%) of preprints.

### Conclusions

More than a quarter of preprints citations were inappropriate since preprints were in fact already published at the time of publication of the citing article, often with a different content.

**Data Availability Statement:** The datasets used and analysed during the current study are available at https://doi.org/10.5281/zenodo.5985960.

**Funding:** The authors received no specific funding for this work.

**Competing interests:** The authors have declared that no competing interests exist.

Authors and editors should check the accuracy of the citations and of the quotations of preprints before publishing manuscripts that cite them.

## Introduction

The scientific community has reacted rapidly to the medical challenges generated by the coronavirus disease 2019 (COVID-19) pandemic, with a steady increase of literature in peer-reviewed journals but also in preprint repositories [1–3]. Yet, a steep increase in the number of posted preprints was observed between January and December 2020, from 797 to 14 290, out of which 8858 (62.0%) were COVID-19-related [4].

However, preprints posted on preprint platforms are not subject to peer-review, and therefore critical appraisal, until they are submitted to peer-reviewed journals. Only a small proportion will be converted to scientific publications and the share of converted preprints shows a declining trend over time [4–7]. For those finally passing the peer-review process, peer-reviewers' comments and critiques lead authors to revise their manuscripts, substantially at times [4, 6, 8]. In that case, the preprint is not the latest version of the work anymore and identifying if it has been published, and referring to the published version instead of the preprint is an important issue as it provides readers with the latest version of a now certified work.

However, matching preprints to subsequent peer review publications is a challenge. Once a preprint is published in a peer-reviewed venue, the preprint server is supposed to update its web page, adding a prominent hyperlink leading to the newly published work. It has been observed that the preprint server MedRxiv reports only 39.7% of all existing publication links [9]. It is therefore probable that some studies are inadequately cited as preprints, although the work cited has been already published, and sometimes with a different content.

The first objective of our study was to assess the reliability of preprint citations in articles on COVID-19 published by the world leading medical journals. The second objective was to assess the rate of publication of preprints cited in articles appearing in theses journals and the last objective was to compare, if relevant, the content of the preprints to their published version.

## Materials and methods

All research articles published on COVID-19 in the first 11 months of 2020 in the British Medical Journal (BMJ), The Lancet, the Journal of the American Medical Association (JAMA) and the New England Journal of Medicine (NEJM) were identified through the journal websites. We then manually screened these articles to identify all articles citing at least one preprint from MedRxiv, in the text as a footnote but also in the reference list since all these journals allow citation of preprints in the reference list of articles they published.

In September 2021, we searched PubMed, Google and Google Scholar to assess if each preprint had been published in a peer-reviewed journal, using the title and the first author in the search string. For each of those which had been published, we compared preprint version and associated journal article to identify changes in the title and two evidence components: study results (*e.g.* numeric changes in sample size, hazard ratio, odds ratio, event rate, or change in p-value) and abstract conclusions (staying positive, negative or neutral regarding the intervention effect, and reporting uncertainty in the findings or not). When different versions of the preprint were available, we used the version existing at the time the citing article was first published online. The results were considered different when the sizes of the samples, the main

quantitative results or the level of significance were different. The conclusions were considered similar if the conclusions in the abstract of the published version remained positive, neutral, or negative regarding the effect of the intervention, or reported the same uncertainties. The conclusions were considered different otherwise. We finally collected the date of first posting of the preprint in MedRxiv, the date of publication of the article citing the preprint and the date of online publication in a journal of the preprint, if relevant.

Data were included in an Excel spreadsheet (Microsoft®).

## Results

The four major medical journals published 205 research articles on COVID until the 1st of December 2020, among which 60 (29.3%) cited at least one preprint from medRxiv (Table 1).

Overall, 182 preprints were cited, among which 124 (68·1%) have been published in a peer-reviewed journal (Fig 1). None of the preprints cited were withdrawn from medRxiv server.

However, among those 124 "preprints", 51 (41·1%) had already been published when the citing article was published online, at least 2 months before for 33 of them, and were, in fact, not preprints anymore at that time.

Among these 51 superseded preprints, differences between the preprint version that was cited and the published version were observed in the data or the conclusions for 21 (40%) of them and in the title for 23 (45%) of them (Table 2).

For the 73 other preprints, *i.e.* the 73 (124–51) which were not already published when the citing article was made available online, differences between the preprint version that was cited and the published version were observed in the data or the conclusions for 33 (45%) of them and in the title for 24 (33%) of them (Table 2).

The differences in the results were often linked to different sample sizes. For example, in the study assessing factors associated with hospital admission and critical illness, the preprint version included 4,103 participants whereas the article published in the BMJ included 5,279 participants [10, 11]. This led to different conclusions, with an odds ratio for hospital admission among people $> 75$ years shifting from 66.8% (95% confidence interval 44.7 to 102.6) in the preprint to 37.9% (95% confidence interval 26.1 to 56.0) in the final article.

If we consider only the preprints that were not already published when the citing article was available online, the rate of publication was 55.7% (73 out of 131), with a median publication delay of 65 days (minimum 1 –maximum 486).

The fact that the preprint had been published in a peer-reviewed journal was not mentioned in medRxiv for 20 (39.2%) of the 51 superseded preprints and 33 (45.2%) of the 73 other preprints ($p = 0.23$).

Overall, the median interval between the first posting of a preprint in MedRxiv and its publication in a peer-reviewed journal was 94 days.

**Table 1. Research articles published on COVID up to the 1st of December 2020, and citation of preprints.**

|  | Number of articles published | Articles citing at least one preprint (%) | Total number of references in articles citing at least one preprint | Number of preprints among the references (%) |
|---|---|---|---|---|
| BMJ | 35 | 20 (57.1%) | 956 | 91 (9.5%) |
| JAMA | 89 | 14 (15.7%) | 402 | 26 (6.5%) |
| Lancet | 37 | 16 (43.2%) | 647 | 53 (8.2%) |
| NEJM | 44 | 10 (22.7%) | 282 | 12 (4.3%) |
| Total | 205 | 60 (29.3%) | 2287 | 182 (8%) |

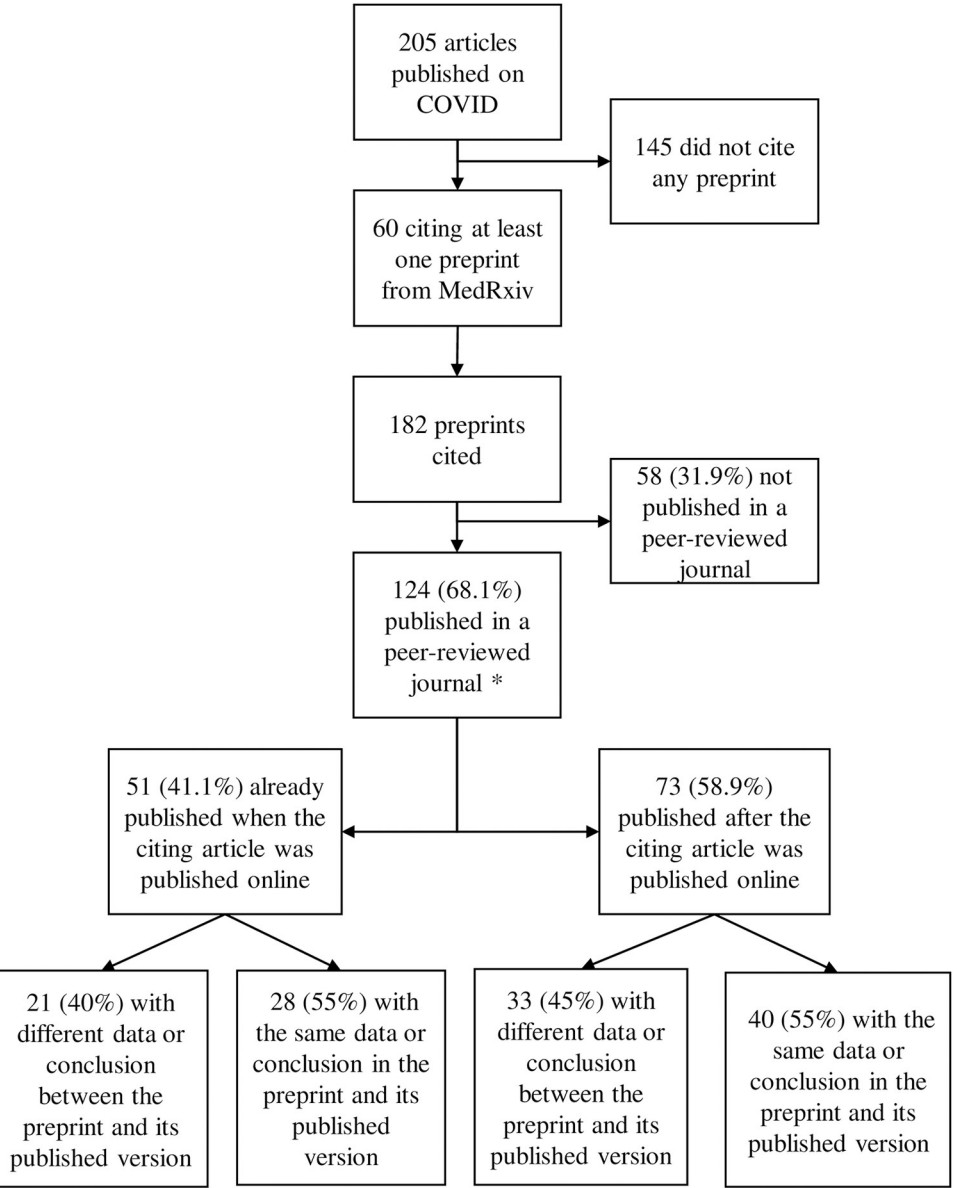

**Fig 1. Flow chart of preprint citations and publications.**

## Discussion

We found that preprints are frequently cited in research articles on COVID-19 published in the world's leading medical journals. However, nearly half of the citations are inappropriate since the preprint was already published when the citing article was published online. Furthermore, many preprints cited will not be published in peer-reviewed journals, or will be but with different data.

Articles on COVID have been published in many different journals but we chose to investigate the BMJ, The Lancet, the JAMA and the NEJM because they are the medical journals that have published much of the research on Covid-19 [12, 13].

**Table 2. Differences between the preprint and its published version.**

|  | SP[a] | P[b] |
|---|---|---|
| **Same data / same conclusion** | 30 (59%) | 40 (55%) |
| **Same data / different conclusion** | 0 (0%) | 0 (0%) |
| **Different data / same conclusion** | 16 (31%) | 20 (27%) |
| **Different data / different conclusion** | 5 (10%) | 13 (18%) |
| **Different title** | 23 (45%) | 24 (33%) |

[a]SP: supersided preprints, preprint published before the publication of the citing article
[b]P: preprint published after the publication of the citing article

Many preprint platforms exist, the oldest coming from the early 1990[th] in the physical sciences. We chose to use MedRxiv, which was launched in June 2019 to provide a dedicated platform and processes for preprints in medicine and health related sciences, because it became particularly popular during the pandemic and is the server that hosted the largest number of preprints on COVID-19 [1, 14].

More than 40% of these citations were inappropriate since, at the time of online publication of the citing article, the preprint was already published in a peer-reviewed journal.

The fact that a large proportion of preprints cited in articles published in peer-reviewed journals were not preprint anymore at the time of publication of the citing article has not been reported in the literature, to our knowledge. This would not be a problem if the contents of the preprint and its published version were similar. However, we observed differences in the data and even a different conclusion for nearly half and 10% of them, respectively. Therefore, searching for a potential peer-reviewed article before citing a paper as a preprint would better reflect the latest evidence available in many cases. However, identifying the subsequent publication of a preprint can be difficult since we observed, that MedRxiv often does not mention the publication of the preprints in a peer-reviewed journal, although it is supposed to display link to journal publication within a month [14], which confirms previous studies [9, 15]. Furthermore, in our study, nearly half of the preprints have a different title than their published version and just a copy and paste in PubMed or Google might not allow identifying the published version.

The interest and validity of preprints have been largely debated, with pros and cons, even before the COVID-19 pandemics [16]. This debate has largely increased about COVID preprints, balancing between the interests of early dissemination of knowledge and the risk of postulated claims without evidence, which will be used by scientists, healthcare workers, and the general public [4, 17, 18].

Preprints posted on platforms are not subject to peer-review and therefore critical appraisal, but some preprint platforms perform screening check, usually related to scope of the article (*e.g.* scientific content, not spam, relevant material, language), plagiarism and legal/ethical/societal issues and compliance [14]. Among the 44 preprint platforms identified by Kirkham *et al.* as having biomedical and medical scope, only three of them (Research Square, bioRxiv and medRxiv) check whether the content contains unfounded medical claims [14].

Despite these checks, MedRxiv platform states that "preprints are preliminary reports of work that have not been certified by peer review. They should not be relied on to guide clinical practice or health-related behavior and should not be reported in news media as established information »

However, preprints have been widely used during the COVID-19 pandemic where much of the preliminary evidence has been made available through preprints at the time of WHO

declaring the epidemic a public health emergency. Only a fraction of these preprints is subsequently published in peer-reviewed journals.

In our study, nearly half of the preprints which had not been published in a peer-reviewed journal before being cited as a preprint, were not subsequently published in peer-reviewed journals.

Our study may have underestimated the real rate of publication of cited preprints. However, among the preprints that were not finally published, all but one had been posted in MedRxiv before mid-August 2020 and we searched for publication more than one year after the posting. A subsequent publication is therefore improbable, considering the usual median delay of publication reported in other studies, ranging from 28 to 110 days [5–7, 19], with a mean of 68 days according to a large review [1]. In our study, all the preprints subsequently published were published within 10 months following the publication of the citing article, but one.

The rate of subsequent publications of preprints cited in the four world-leading medical journals is higher than the rate of publication of preprints reported in other studies, ranging from 6.9 to 21.1% [1, 4, 5, 7, 19–21]. However, the rates of publication reported in those studies are probably underestimated since they relied on information about subsequent publication in MedRxiv, which we observed to be largely inaccurate, confirming previous reports [9].

For the preprints which were finally published, the differences between the preprints and their published version were frequent and close to those reported in another study on 139 preprints, with changes in the study results or the abstract conclusions between the first preprint version to journal article for 38% and 29% of preprints, respectively [6]. However, when studying 67 studies on Covid-19 prevention or treatment, Bero *et al.* found only 23 articles (34%) that had no discrepancies in results reporting between preprints and journal publications [8].

The changes in the conclusions were less frequent than the changes in studies' result, mostly because conclusions were usually broad, such as "among patients hospitalized with Covid-19, those who received hydroxychloroquine did not have a lower incidence of death at 28 days than those who received usual care" [22].

These changes can be the consequence of publishing preliminary results in a preprint and final results in a peer-reviewed journal, of the fact that some preprints receive comments that allow authors to improve their manuscript prior to submission to a journal, or of the improvement of the quality of the report following the peer reviewed process.

Preprints on COVID-19 posted in preprint servers are shorter and contain less references than non-COVID-19 preprints, which suggests that authors tend to publish preliminary results [1].

Concerning gradual quality improving, although preprint servers offer authors the opportunity to post updated versions of a preprint, enabling them to incorporate feedback, correct mistakes, or add additional data and analysis, the majority of preprints on COVID-19 in preprint servers exist in fact as only a single version [1]. A study on bioRxiv preprints showed that less than 10% of preprints received at least one comment, and one third were posted by the preprint's authors themselves [23].

Although little empirical evidence is available to support the use of editorial peer review as mechanism to ensure quality of biomedical research [24], the peer review process is a still considered a cornerstone to improve the quality of scientific publications [25].

In light of that, in our study, when combining the 58 preprints that were published and the 56 that appeared finally in a peer-reviewed journal, but with different results or conclusions, we can consider that 62.6% of the quotations of preprints may not be fully reliable.

Another interest of preprints is that they allow free access to research findings, while a large proportion of journal articles often remain behind subscription paywalls. In response to the

pandemic, a number of journal publishers began to alter their open-access policies in relation to COVID-19 manuscripts and made COVID-19 literature temporarily open access [1].

Finally, the quality of research had to be, in response to the pandemic, put in balance with the need to get rapidly new information to tackle this new threat. Communicating science through preprints allow to share research at a faster rate and with greater transparency than allowed by the current journal infrastructure. Nevertheless, according to the shortcomings of preprints, they should not appear in the reference list of journals but could be cited in the text as such, or, at least, the preprint status should be made clear in the text of a manuscript if citing the research and not just in the reference list [25, 26].

A promising alternative is the introduction of brief summaries of preprints with significant potential to impact clinical practice in a specific section of biomedical journals [27, 28]. Another one is to replace the word "preprint", which can be misleading, by 'Unrefereed manuscript', "Manuscript awaiting peer review" or "Non-reviewed manuscript" [29].

## Conclusions

More than 40% of preprints cited in the major medical journals were in fact already published at the time of publication of the citing article. Therefore, authors should check the accuracy of the citation and the quotations of preprints just before submitting the manuscript, and once again when signing the proofs. They should not rely only on the information displayed on MedRxiv website to identify subsequent publication. To overcome medRxiv weaknesses, authors should manually search bibliographic databases to determine if a preprint has been subsequently peer-reviewed and published, keeping in mind the nearly half of the preprints have a different title than their published version. Publishers should also check the accuracy of the citations of preprints before online publication of citing articles.

The debate on the interest of preprints has largely increased during the COVID preprints, balancing between the interests of early dissemination of knowledge and the risk of postulated claims without evidence. However, considering the significant number of changes in the content or even the conclusions between the preprints and their published version, quotations of preprints should be considered with caution, even in the articles published in the major medical journals.

## Supporting information

**S1 File.** https://doi.org/10.5281/zenodo.5985960.
(XLSX)

## Acknowledgments

The authors thank Mrs Anna Gehanno for her help in editing the manuscript.

## Author Contributions

**Conceptualization:** Jean-Francois Gehanno, Laetitia Rollin.

**Data curation:** Jean-Francois Gehanno, Laetitia Rollin.

**Formal analysis:** Jean-Francois Gehanno, Julien Grosjean, Stefan J. Darmoni, Laetitia Rollin.

**Supervision:** Laetitia Rollin.

**Validation:** Jean-Francois Gehanno.

**Writing – original draft:** Jean-Francois Gehanno, Laetitia Rollin.

**Writing – review & editing:** Jean-Francois Gehanno, Julien Grosjean, Stefan J. Darmoni, Laetitia Rollin.

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
