## [Decision Letter · Decision Letter 0]

9 Jun 2022

PONE-D-22-04047Reliability of citations of medRxiv preprints in articles published on COVID-19 in the world leading medical journalsPLOS ONE

Dear Dr. Gehanno,

Thank you for submitting your manuscript to PLOS ONE. After careful consideration, we feel that it has merit but does not fully meet PLOS ONE’s publication criteria as it currently stands. Therefore, we invite you to submit a revised version of the manuscript that addresses the points raised during the review process.

We look forward to receiving your revised manuscript.

Kind regards,

Venkatesh Shankar Madhugiri

Academic Editor

PLOS ONE

Journal Requirements:

Additional Editor Comments:

Please make edits in the paper per the reviewer's comments.

Reviewers' comments:

Reviewer's Responses to Questions

**Comments to the Author**

1. Is the manuscript technically sound, and do the data support the conclusions?

Reviewer #1: Partly

2. Has the statistical analysis been performed appropriately and rigorously? 

Reviewer #1: Yes

3. Have the authors made all data underlying the findings in their manuscript fully available?

Reviewer #1: Yes

4. Is the manuscript presented in an intelligible fashion and written in standard English?

Reviewer #1: Yes

5. Review Comments to the Author

Reviewer #1: I agree with the other reviewers that this could go as a commentary rather than a full research article. Since the analysis is not complex and does not involve sufficient examples, for example across multiple platforms, the conclusions reached cannot be generalized. At the same time, the authors have drawn attention to an interesting problem arising from the citation of non-peer reviewed work in published papers. The authors have also mentioned some cautionary measures.

I believe this paper should be published.

Some minor changes in language ->

71 May like to use “important issue” in place of “major…”….

In the following, X denotes crossed out.

73 Yet …Remove

75 Nevertheless…Remove

85 11 months? …or should it be 31 December?

113 Overall , 182 UNIQUE preprints….?

234 a Remove

248 shortcoming -> shortcomings

6. PLOS authors have the option to publish the peer review history of their article (what does this mean?). If published, this will include your full peer review and any attached files.

Reviewer #1: No

---

## [Author Response · Author response to Decision Letter 0]

4 Jul 2022

Reviewers' comments:

Reviewer #1: I agree with the other reviewers that this could go as a commentary rather than a full research article. Since the analysis is not complex and does not involve sufficient examples, for example across multiple platforms, the conclusions reached cannot be generalized. At the same time, the authors have drawn attention to an interesting problem arising from the citation of non-peer reviewed work in published papers. The authors have also mentioned some cautionary measures.

I believe this paper should be published.

Some minor changes in language ->

71 May like to use “important issue” in place of “major…”….

In the following, X denotes crossed out.

73 Yet …Remove

75 Nevertheless…Remove

85 11 months? …or should it be 31 December?

113 Overall , 182 UNIQUE preprints….?

234 a Remove

248 shortcoming -> shortcomings

We thank reviewer #1 for his general comments on the manuscript and for the changes in language he suggested.

All the changes have been made in the revised manuscript.

---

## [Editor Report · Decision Letter 1]

26 Jul 2022

Reliability of citations of medRxiv preprints in articles published on COVID-19 in the world leading medical journals

PONE-D-22-04047R1

Dear Dr. Gehanno,

We’re pleased to inform you that your manuscript has been judged scientifically suitable for publication and will be formally accepted for publication once it meets all outstanding technical requirements.

Kind regards,

Venkatesh Shankar Madhugiri

Academic Editor

PLOS ONE
---

## [Editor Report · Acceptance letter]

29 Jul 2022

PONE-D-22-04047R1 

Reliability of citations of medRxiv preprints in articles published on COVID-19 in the world leading medical journals 

Dear Dr. Gehanno:

I'm pleased to inform you that your manuscript has been deemed suitable for publication in PLOS ONE. Congratulations! Your manuscript is now with our production department. 

Kind regards, 

on behalf of

Dr. Venkatesh Shankar Madhugiri 

Academic Editor

PLOS ONE